# BEYOND GAMES: BRINGING EXPLORATION TO ROBOTS IN REAL-WORLD

## ABSTRACT

Exploration has been a long standing problem in both model-based and model-free learning methods for sensorimotor control. While there has been major advances over the years, most of these successes have been demonstrated in either video games or simulation environments. This is primarily because the rewards (even the intrinsic ones) are non-differentiable since they are function of the environment (which is a black-box). In this paper, we focus on the policy optimization aspect of the intrinsic reward function. Specifically, by using a local approximation, we formulate intrinsic reward as a differentiable function so as to perform policy optimization using likelihood maximization – much like supervised learning instead of reinforcement learning. This leads to a significantly sample efficient exploration policy. Our experiments clearly show that our approach outperforms both on-policy and off-policy optimization approaches like REINFORCE and DQN respectively. But most importantly, we are able to implement an exploration policy on a robot which learns to interact with objects completely from scratch just using data collected via the differentiable exploration module. See project videos at `https://doubleblindICLR.github.io/robot-exploration/`.

## 1 INTRODUCTION

There has been a lot of recent progress in the field of Reinforcement Learning (RL). However, most of the successful applications have been confined to the artificial world of video games (Mnih et al., 2015b) or simulations (Lillicrap et al., 2016). While the field of RL was born out of need to make our robots learn how to perform actions, none of the recent advances have translated to success in the field of robotics. Why is that? Let us consider the simple task of stacking. How does the robot learn to execute successful trajectories for stacking? In model-free Reinforcement Learning(RL) paradigm, the robot will try and try until it is able to stack objects and once it hits a successful instance, it is used as a positive signal ('reward') to promote these policy parameters. How does the robot try? Due to lack of any other signals from the environment, most-often robots use random-exploration policies (or random trajectories). It is clear that if the rewards are sparse, it may take millions of random "tries" before it hits any success. Clearly this approach is only scalable in video-games and not real-world robotics. Another possibility is to use model-driven approaches. Here, the robot will learn a model of the world from our millions of interactions and use the model to simulate and search. But what millions of interactions should be performed to build our models? Again due to lack of any external information, the most common approach is using random interactions to explore and build the world model (Agrawal et al., 2016; Levine et al., 2016; Pinto et al., 2016). Building a good model will require enormous number of interactions.

It is clear that one of the biggest stumbling blocks in-front of current robotics is lack of a structured way to explore the world and be efficient in their tries to seek reward or build a model. Therefore, there has been a lot of significant effort to build approaches for exploration and being more sample-efficient than random or on-policy exploration. The common theme across these approaches is to introduce "intrinsic" rewards – rewards given by agent to itself based on how environment behaves. These rewards are denser compared to external rewards and hence provide early feedback to the exploration policy. Some examples of intrinsic rewards include "curiosity" (Pathak et al., 2017; Oudeyer & Kaplan, 2009; Schmidhuber, 1991) where prediction error is used as reward signal, "diversity rewards" (Eysenbach et al., 2018; Lehman & Stanley, 2011b;a) which discourages the agent from revisiting the same states (or similar states). If the intrinsic rewards are treated as additional

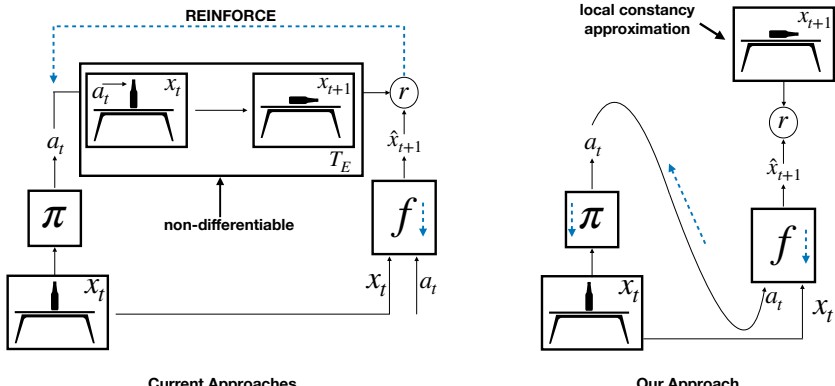

Figure 1: Schematic Explanation: In general formulation, the reward function $r$ is a function of environment function $T_{\mathcal{E}}$. This makes reward function non-differentiable with respect to action $a_t$. In our approach, we use local constancy approximation to assume $x_{t+1}$ is a constant. This immediately turns reward function differentiable and hence trainable via gradient descent.

rewards in RL and explored in context of original extrinsic rewards; it acts as reward shaping function which improves sample efficiency but not by a significant amount.

Another possibility is to use the inspiration from humans: humans even try to explore the world without the context of task. In a similar-manner, we can use these intrinsic rewards to learn task-independent exploration policy. This task-independent exploration policy can then be fine-tuned with sparse task-rewards or used to collect data to train model-based RL algorithms. In fact, in a recent work (Burda et al., 2018), the authors demonstrate how curiosity can be used to train exploration-policy and then fine-tuned for specific tasks. And they demonstrate the power of curiosity on $54$ games. Yes, 54 environments but no real-world physical robots!

Why is that? To understand the reason behind sample inefficiency of curiosity or intrinsic rewards, notice how the intrinsic rewards are given by agent. The agent performs an action and then computes the reward based on its own model and environment behavior. For example, in curiosity (Pathak et al., 2017), if the internal model and environment behavior disagree, then the policy is rewarded. From an exploration viewpoint this seems like a good formulation, rewarding actions for which model knows little-to-none. But the same formulation from an optimization viewpoint, it suffers from all the bad properties of extrinsic rewards. The reward is a function of environment behavior with respect to the performed action. Since the environment behavior function is unknown, it is treated as black-box and hence the gradients have to be computed using REINFORCE (Williams, 1992) which is quite sample inefficient.

Our paper investigates exploration from an optimization viewpoint and asks a simple question: can we formulate curiosity reward as a differentiable function? We believe a differentiable reward function would enable to us to be sample efficient and for the first time ever, implement exploration on a real-world physical robot. We use a simple yet quite an effective approximation which yields the reward function differentiable. In the end, our optimization is simple min-max over parameters of forward model and policy function results. Our results indicate that the differentiable reward function formulation is significantly better in exploration reaching to interesting interactions in few hundred tries. This allows us to implement an exploration policy on a robot and demonstrate pushing and grasping actions using policies trained from scratch.

## 2 INCENTIVIZING EXPLORATION VIA INTRINSIC REWARDS

Consider an agent interacting with the environment $\mathcal{E}$. At time $t$, it receives the observation $x_t$ and then takes an action predicted by its policy, i.e., $a_t = \pi(x_t; \theta_P)$. Upon executing the action, it receives, in return, the next observation $x_{t+1}$ which is 'generated' by the environment. Our goal is to build an agent that chooses its action in order to maximally explore the state space of the environment in an efficient manner. A popular way to train an agent to perform efficient exploration is to incentivize it by giving feedback on the samples it already executed in the environment via "intrinsic" rewards, i.e., the rewards generated by the agent itself.

**Strategies for Intrinsic Reward Generation:** Given the agent's transition $\{x_t, a_t, x_{t+1}\}$, let the intrinsic reward generated by the agent be $r^i(x_t, a_t, x_{t+1})$ abbvt. as $r_t^i$. A good intrinsic reward formulation would be the one that encourages the agent to perform actions that lead to most informative examples. In general, the reward function is considered black-box since it involves environment. In this paper, we demonstrate that a simple gradient approximation for maximizing rewards which are intrinsic to the agent leads to a much more sample-efficient policy optimization procedure. While our formulation could be applied to a more general set of intrinsic rewards, we describe it in the context of prediction-error based curiosity rewards which has recently been shown to be successful across a large variety of simulated environments (Burda et al., 2018).

The prediction-error based curiosity reward formulation (Pathak et al., 2017) involves first building a prediction model, aka forward model, $f_{\theta_F}$ of the environment. Forward model is trained to map the current observation $x_t$ and $a_t$ to the resulting state $x_{t+1}$ using maximum likelihood loss and hence it can be learned efficiently. The agent's policy is trained to select the actions $a_t$ which result in high loss for the forward model. For instance, in deterministic environment, a gaussian density model can be used to define intrinsic reward $r_t^i$ for the agent:

$$r^i(x_t, a_t, x_{t+1}) \triangleq \|f(x_t, a_t; \theta_F) - x_{t+1}\|_2 \tag{1}$$

The agent is trained to maximize $r_t^i$ and the forward model is simultaneously trained in online manner on the data collected by the agent during exploration. One commonality among different exploration methods (Bellemare et al., 2016; Pathak et al., 2017; Houthooft et al., 2016a), is that the forward model is usually learned in a supervised manner and the agent's policy is trained using reinforcement learning either in on-policy or off-policy manner.

**On-Policy Reward Optimization**  In this case, the policy is directly optimized to maximize the intrinsic reward $r_t^i$ via policy-gradients (Sutton & Barto, 1998). Given the agent's rollout sequence and the intrinsic reward $r_t^i$ at each timestep $t$, the policy is trained to maximize the sum of expected reward, i.e., $\max_{\theta_P} \mathbb{E}_{\pi(x_t; \theta_P)} \left[ \sum_t \gamma^t r_t^i \right]$ discounted by a factor $\gamma$. The gradients of this expression are computed using REINFORCE (Williams, 1992). In practice, the existing on-policy algorithms, e.g., A3C (Mnih et al., 2016), PPO (Schulman et al., 2017) etc. are deployed off-the shelf.

**Off-Policy Reward Optimization**  Off-policy algorithms (e.g. DDPG (Lillicrap et al., 2016) or DQN (Mnih et al., 2015a)) use Q-value function to represent agent's policy. Q-value, for a given pair of the current state $x_t$ and the executed action $a_t$, is defined as the total sum of discounted future rewards. Agent's policy can be written in terms of this Q-value function as $a_t = \pi(x_t; \theta_P) = \arg\max_{a_t} Q(x_t, a_t; \theta_P)$ plus epsilon-noise. The main benefit in learning Q-values is that they can be trained with samples from buffer even if they are not from the agent's current policy to minimize the following loss: $\|Q(x_t, a_t; \theta_P) - \mathbb{E}_{\pi(x_t; \theta_P)} \left[ \sum_t \gamma^t r_t^i \right]\|_2$. However, since intrinsic reward distribution $r_t^i$ changes over time one would need to update the intrinsic reward using the most recent $f_{\theta_F}$.

The key thing to note is that both the approaches to policy optimization treat reward $r_t^i$ as an unknown quantity which can only be estimated via samples.

## 3    SAMPLE-EFFICIENT DIFFERENTIABLE EXPLORATION

As discussed in the previous section, there has been a lot of work in proposing formulations for intrinsic rewards to the agent. However, the optimization procedure for training policies to maximize these intrinsic rewards has more or less remained the same – i.e. – treating the intrinsic reward as a "black-box" even though it is generated by the agent itself.

Let's consider an example to understand the reason behind the status quo. Consider a robotic-arm agent trying to push multiple objects kept on the table in front of it by looking at the image from an overhead camera. Suppose the arm pushes an object such that it collides with another one on the table. The resulting image observation following this action will be the outcome of complex real-world interaction the actual dynamics of which is not known to the agent. More importantly, note that this resulting image observation is a function of the agent's action (i.e., push in this case). Since the intrinsic reward $r^i(x_t, a_t, x_{t+1})$ is function of the next state (which is a function of the agent's action) . This dependency on the unknown environment dynamics absolves the policy optimization to compute any sort of analytical gradient with respect to the intrinsic rewards.

To state this intuition mathematically, let $T_{\mathcal{E}}$ be the true transition function of the environment such that $x_{t+1} = T_{\mathcal{E}}(x_t, a_t)$. Note that for environments with stochastic dynamics $T_{\mathcal{E}}$ will be a distribution, but we describe deterministic case for brevity of the notations. The intrinsic reward function $r_t^i$, in Equation (1), can now be rephrased in terms of $T_{\mathcal{E}}$ as $r^i(x_t, a_t, T_{\mathcal{E}}(x_t, a_t))$. It is easy to see that $T_{\mathcal{E}}$ is required to compute analytical gradients $\partial r_t^i / \partial a_t$ at any time $t$.

The standard way to optimize policy to maximize sequence of intrinsic rewards is to either use REINFORCE (i.e., on-policy way) or regress to $r_t^i$ to learn value estimates (i.e., off-policy) as discussed in the previous section. Both these approaches do not make any use of the structure present in the design of $r_t^i$. While these are unbiased estimators for training policy parameters with respect to $r_t^i$, they suffer from high variance which is a known issue in reinforcement learning and an active area of research (Schulman et al., 2015). For instance, REINFORCE roughly amounts to saying that the agent should change the probability of the executed action in proportion to rewards received which fluctuates with the reward trajectories, leading to high variance. It gives no signal as to what action to take if the current action did not lead to a good reward.

### 3.1 DIFFERENTIABLE INTRINSIC REWARD WITH LOCAL CONSTANCY APPROXIMATION

The focus of this paper is on the policy optimization aspect of the intrinsic reward function rather than their formulation. Our goal is to address the question whether we can formulate intrinsic reward as a differentiable function so as to perform policy optimization using likelihood maximization – much like supervised learning instead of reinforcement. If possible, this would allow the agent to make use of the structure in $r_t^i$ explicitly. For instance, in case of curiosity-driven intrinsic reward, the forward predictor $f_{\theta_F}$ is trained via maximum likelihood which means it can be learned with much more sample efficiency. If the policy could also be optimized using direct gradients, the rewarder could very efficiently inform the agent to change its action space in the direction where forward prediction loss is high, instead of providing a *scalar* feedback as in case of reinforcement learning. Explicit reward (cost) functions are one of the key reasons for success stories in optimal-control based robotics (Deisenroth & Rasmussen, 2011b; Gal et al., 2016), but they don't scale to high-dimensional state space such as images and rely on having access to a good model of the environment.

To address our goal, let us revisit the intrinsic reward function in Equation (1). Upon substituting $\pi_{\theta_P}$ and $T_{\mathcal{E}}$, we get:

$$r_t^i \triangleq \| f(x_t, \pi(x_t; \theta_P); \theta_F) - T_{\mathcal{E}}(x_t, \pi(x_t; \theta_P)) \|_2 \tag{2}$$

Note that the first term in the Equation (2) is differentiable since $f_{\theta_F}$ is a learned function, but second term is not since the environment is not known. Hence, we propose to make a local zeroth-order approximation to the environment so as to make $r_t^i$ differentiable with respect to the action $a_t$.

**Local constancy approximation** Consider a state $x_t$ and the action $a_t$ predicted by the policy at that state, $a_t = \pi(x_t; \theta_P)$. The agent reaches $x_{t+1}$ upon executing the action. We assume that the final state $x_{t+1}$ remains constant in an infinitely small $\epsilon$-neighborhood ball around the action $a_t$ executed at the same initial state $x_t$. Mathematically, it can be written as $x_{t+1} = T_{\mathcal{E}}(x_t, a_t) \approx T_{\mathcal{E}}(x_t, \tilde{a}_t)$ for $\tilde{a}_t \in N_\epsilon(a_t)$ where $\epsilon \to 0$. With this approximation in place, we can now compute $\partial r_t^i / \partial a_t$ at any time $t$ by back propagating gradients through the forward function.

Intuitively, it basically assumes that minute fluctuations in action won't change the transition state. Of course, this does not hold true in certain corner cases, for instance, pushing an object at the very corner edge. In those cases, intrinsic reward formulation with local constancy approximation would incentivize all the actions equally in the $\epsilon$-neighborhood ball, instead of that particular action. This, fortunately, seems like a reasonable settlement in practice as such corner cases require higher sampling to understand where exactly the corner case occurs.

### 3.2 DIRECT POLICY OPTIMIZATION WITH DIFFERENTIABLE INTRINSIC REWARDS

We now leverage the local constancy approximation to formulate a direct gradient-based policy optimization which maximizes intrinsic rewards achieved by the agent without relying on reinforcement learning for policy optimization.

We first discuss the one step case and then provide the general setup. Given a transition $\{x_t, a_t, x_{t+1}\}$, the agent generates an intrinsic reward for itself $r_t^i = \| f(x_t, a_t; \theta_F) - x_{t+1} \|_2$. Forward model $f_{\theta_F}$

is trained to minimize its loss which amounts to minimizing $r_t^i$ with respect to $\theta_F$. This is done via direct loss minimization using gradient descent. Upon using *local constancy approximation*, we can also optimize for policy parameters $\theta_P$ via differentiable loss function in the same manner using gradient descent. However, policy is optimized to maximize the objective. This results into a min-max optimization over $r_t^i$. The joint objective for a one-step reward horizon is written as follows:

$$\min_{\theta_F} \ \max_{\theta_P} \ \|f(x_t, a_t; \theta_F) - x_{t+1}\|_2 \tag{3}$$
$$\text{s.t.} \quad a_t = \pi(x_t; \theta_P)$$
$$x_{t+1} = T_{\mathcal{E}}(x_t, a_t) \approx T_{\mathcal{E}}(x_t, \tilde{a}_t) \quad \text{for} \quad \tilde{a}_t \in N_\epsilon(a_t), \epsilon \to 0$$

Note that both policy and forward predictor are trained via maximum likelihood in a supervised manner. Hence, given the local constancy approximation, this should be a much more efficient way to optimize exploration policy unlike reinforcement learning based policy-optimization. We optimize the objective in Equation 3 in an alternating fashion where forward predictor is optimized in the outer-loop, and the policy in inner-loop.

**Generalization to multi-step reward horizon** To optimize policy to maximize a discounted sum of sequence of future intrinsic rewards $r_t^i$ in a differentiable manner, the forward predictor would have to make predictions spanning over multiple time-steps. The objective from Equation (3) can be generalized to the multi-step horizon setup by recursively applying the forward predictor as $\sum_t \|\hat{x}_{t+1} - x_{t+1}\|_2$ where $\hat{x}_{t+1} = f(\hat{x}_t, a_t; \theta_F)$ and $\hat{x}_0 = x_0$. Alternatively, one could use LSTM to make forward model itself multi-step.

### 3.3 Practical Considerations in Implementation

**Min-max optimization** At first glance, it might appear that optimizing the objective in Equation (3) could be unstable learning process. However, unlike the other cases of online min-max optimization (e.g. Generative Adversarial Networks (Goodfellow et al., 2014)), this case could be easily made stable. Here, the goal of our policy optimization is to learn from the forward predictor in the outer-loop and the forward predictor is to improve itself. Hence, we train the forward predictor slightly faster than the policy by keeping higher learning rate to stabilize the learning process. Other alternative could be to take few extra gradient steps in the outer-loop minimization than inner-loop maximization.

**Back-propagation through forward predictor** To directly optimize the policy with respect to the loss function of the forward predictor, we need to backpropagate all the way through action sampling process from the policy. In case of continuous action space, one could achieve this via making policy deterministic, i.e. $a_t = \pi_{\theta_P}$ with epsilon greedy sampling (Lillicrap et al., 2016). Alternatively, in case of discrete action space, we found that straight-through estimator (Bengio et al., 2013) works well in practice. In this paper, we discretized the action space of our agent.

**Learning forward predictions in the feature space** It has been shown that learning forward-dynamics predictor $f_{\theta_P}$ (Burda et al., 2018; Pathak et al., 2017) in some feature space leads to better generalization instead of making predictions in raw pixel space. Our formulation is trivially extensible to any representation space because all the operations can be performed with $\phi(x_t)$ instead of $x_t$.

## 4 Experimental Setup and Baselines

We consider the task of object manipulation in complex scenarios. Our setup contains a 7-DOF robotic arm which could be tasked to interact with the the objects kept on the table in front of it. The objects are kept randomly in the workspace of the robot on the table. All of our experiments use raw visual RGBD images as input and predict raw actions as output. We use position-control for controlling the robotic arm and its action space contains following actions: (a) $\{X, Y\}$: the target location in the work-space of the end-effector; (b) $\Theta$: the angle at which gripper of the robot should approach the specified location; (c) Gripper Status: a boolean value indicating whether to perform a grasping (open the gripper fingers) or pushing gesture (keep fingers close). Note that in order to perform an accurate grasp or push on objects, the agent has to figure out accurate location, orientation and the gripper status.

We parametrize the action space in the network in a similar manner as (Zeng et al., 2018). The action space is discretized into $224 \times 224$ $\{X, Y\}$ locations, 16 orientations for grasping (fingers close) and 16 orientations for pushing. Input to the policy is a $224 \times 224$ RGBD image and it produces probabilities of the push and grasp action for each input pixel location. Instead of adding the 16 rotations in the output, we pass 16 equally spaced rotated inputs to the network and then sample actions based on the output of all the inputs. This exploits the convolutional structure of the network.

There is no assumption of any sort about either the environment, or the training signal. Our robotic agents explore the work-space purely out of their own intrinsic reward in a pursuit to develop useful skills. We have an instantiation of this setup both in the real world and in a simulated environment.

**Baselines** In this paper, we propose a sample efficient way to optimize policy using intrinsic rewards that does not treat the reward function as a black box. As baselines, we compare the same curiosity reward function but optimized using REINFORCE. We also compare to off-policy baseline of DQN. In order to make apples-to-apples comparison between the proposed differentiable learning optimization vs. existing black-box optimization, We kept exactly the same setup for all the methods and only changed the optimization procedure. For obtaining the right hyper-parameters we use REINFORCE with external touch-rewards (See Appendix A). As one can see the REINFORCE baseline is quite effective with external rewards.

## 5 EXPERIMENTS

Our goal is to demonstrate a exploration formulation which is sample-efficient enough to be applicable in real-world robotic setups. The two main components of our proposed methodology are the exploration policy and the forward prediction model. We evaluate both the components on object manipulation tasks: (a) in a simulated V-REP based environment, and (b) in real-world robotic setup using Sawyer robot arm. We perform simulation experiments to help us setup the right parameters and do extensive comparisons against REINFORCE and DQN. We consider two environment setups: sparse and non-sparse. In the non-sparse setup, the environment contains multiple objects which makes it easier for the policy to stumble upon the object. In the sparse setup, the work-space of the robot only contains one object.

### 5.1 OBJECT MANIPULATION IN SIMULATION

Our main goal is to deploy the exploration policy in real-world, but we first begin by studying in depth the performance of our proposed approach in contrast to prior formulations that treat intrinsic reward function as a black-box.

**Simulated Object Interaction Setup** We used V-REP simulator to simulate the robot performing grasping and pushing on table top environment. This setup is based on (Zeng et al., 2018). It consist of UR5 robot arm with an RG2 gripper. Dynamics is simulated using Bullet Physics 2.83 physics engine. V-REPs internal inverse kinematics module is used robot motion planning. The objects used in these simulations include 6 3D toy blocks of different shapes, colors of which are randomly chosen during experiments. Perception data is captured using a statically mounted 3D camera in the environment. It provides RGBD images (640x480), without any noise added for depth or color.

**Evaluation of the exploration policy learned by the agent** How do we evaluate if our exploration policy is taking interesting steps? In our setup, one attribute that correlates with interesting-ness is performing actions on object. We use this as a metric to explore how quickly our policy learns to explore interesting part of space. Figures 2a and 2b show the performance when the environment consist of single and multiple objects respectively. It is clear that both REINFORCE and DQN perform quite poorly and do not show any significant improvement even after 3K interactions. On the other hand, our approach is able to exploit the structure in the loss to learn significantly faster. and achieves 40% interaction rate even after 3K interactions. Table 2 shows the interaction results in multi-step setting after first 250 interactions. Again, our multi-step formulation is more effective compared to baseline approaches.

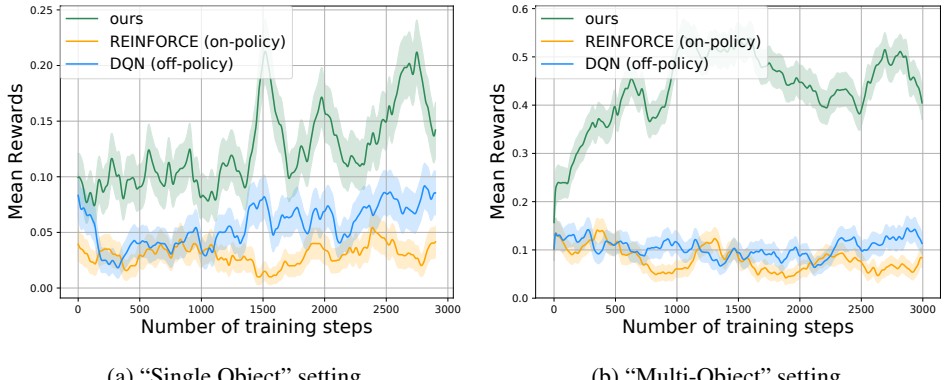

(a) "Single Object" setting          (b) "Multi-Object" setting

Figure 2: Comparing the performance of our proposed sample efficient exploration with other baseline methods in simulated manipulation environment.

## 5.2 OBJECT MANIPULATION USING ROBOTIC ARM IN THE REAL-WORLD

We now deploy our sample-efficient exploration formulation on real-world robotic setup. The real-world poses additional challenges unlike simulated environments in terms of behavior, dynamics of varied objects available in the real world. Our robotic setup consisted of a Sawyer-arm with a table placed in front of it. We mounted Kinectv2 at a fixed location from robot to receive RGBD observation of the environment. We calibrated the system to get extrinsic between camera and robot. In every run robot starts with 3 object placed in front of it (See Appendix for setup). Objects were manually replaced if robot has completed 100 interactions or if there are no objects in front of it. To see if the robot has interacted with objects, we used to monitor the change in the RGB image. This information is only used for our purpose to check the progress of robot, it is not provided to policy.

In order to test the skills learned by the robot during its curious exploration, we tested it on a set of held-out objects. Out of total of 30 objects, we created set of 20 objects for training and 10 objects for testing. Both, our method and reinforce were trained for 1400 robot interaction with the environment. Both models were evaluated based on the 80 robot interaction. During testing , environment reset was done after every 10 robot steps.

**How good is Exploration Policy?** The key requirement of a good exploration policy is that it should search the space in an efficient manner, generalize to unseen scenarios and discover complex behaviors which are hard to stumble upon randomly. Again we use the same metric (interaction with objects) as before to measure effectiveness of exploration policy. Figure 3(left) shows how effective our differentiable curiosity module is and how it learns to interact with object even with 1400 examples. This result clearly indicates the importance of using the approximation and hence differentiable reward function. At the end of 1400 steps, the interaction rate was more than 80%. Our final trained exploration policy interacts approximately 91% of times with unseen objects whereas random performs 17%. On the other hand, it seems that REINFORCE just collapses and only 1% of actions involve interaction with object (See Table 3).

**How good is Forward Prediction Model?** We use the data collected during the exploration to train forward prediction models. If the data explored by the agent is interesting, the prediction model should perform well on complex tasks. We evaluate how well is the forward prediction model in terms of planning for goal-driven scenarios. We also provide the evaluation in terms of future prediction (Finn et al., 2017) in Appendix. We compare the planning accuracy of the forward model learned on the data collected by different exploration schemes. We use the cross-entropy based method to optimize for the action plan given the initial state and the goal state. Metric is L2 distance from the ground truth action location. As result in the Table 1 indicate the forward model learned using our approach is significantly better than the baseline REINFORCE model.

## 6 RELATED WORK

The problem of exploration is a well-studied problem in the field of reinforcement learning. Early approaches focused on studying exploration from theoretical perspective (Strehl & Littman, 2008) and proposed Bayesian formulations (Kolter & Ng, 2009; Deisenroth & Rasmussen, 2011a) which are

| Method Name | Pushing | Failed Grasp Attempt | Grasping | Mean |
|---|---|---|---|---|
| Exploration w/ REINFORCE | 49.22 | 54.97 | 55.49 | 53.22 |
| Exploration w/ Ours | 35.41 | 38.06 | 53.26 | 42.24 |

Table 1: **Planning Error** of the forward prediction learned on the data collected by the exploration policy for multi-object scenarios.

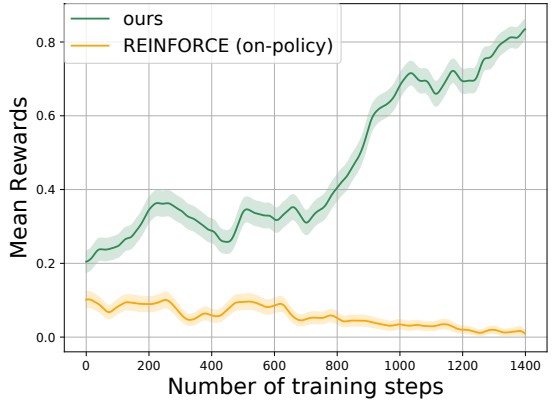

| Table 2: Multi-step Learning (Simulation) | | |
|---|---|---|
| **Ours** | **REINFORCE** | **DQN** |
| **11%** | **6.0%** | **6.5%** |

| Table 3: Real-world Robot Test | | |
|---|---|---|
| **Ours** | **REINFORCE** | **RANDOM** |
| **91%** | **1%** | **17%** |

Figure 3: (Left) Interaction Rate vs. Number of Samples for real-world robot. (Right) Table 2 shows interaction rate with 250 steps and Table 3 shows the interaction rate of final robot model.

hard to scale in higher dimensions. In this paper, we focus on the specific problem of exploration using intrinsic rewards. A large family of approaches use "curiosity" as a reward for training the agents. A good summary of early work in curiosity-driven rewards can be found in (Oudeyer et al., 2007; Oudeyer & Kaplan, 2009). Most approaches use some form of prediction-error between the learned model and environment behavior (Pathak et al., 2017). This prediction error can also be formulated as surprise (Schmidhuber, 1991; Achiam & Sastry, 2017; Sun et al., 2011). Other forms of curiosity can be to explore states and actions where prediction of a forward model is highly-uncertain (Still & Precup, 2012; Houthooft et al., 2016b). Finally, approaches such as (Lopes et al., 2012) try to explore state space which help improve the prediction model most. However, most of these efforts have still studied the problem in context of external rewards. These intrinsic rewards just guide the search to the space where forward model is uncertain or likely to be wrong.

Another approach for intrinsic rewards is using explicit visitation counts (Bellemare et al., 2016; Fu et al., 2017). These exploration strategies guide the exploration policy to "novel" states (Bellemare et al., 2016). A closely related work uses diversity as a reward for exploration and skill-learning (Eysenbach et al., 2018). However, both visitation counts or measuring diversity requires learning a model which keeps the distribution of visited states. Learning such a model does not seem trivial. Another issue is the transferable properties and generalization of such approaches unless the state features are transferable themselves.

Finally, apart from intrinsic rewards, other approaches include using an adversarial game (Sukhbaatar et al., 2018) where one agent gives the goal states and hence guiding exploration. Gregor et al. (2017) introduce a formulation of empowerment where agent prefers to go to states where it expects it will achieve the most control after learning. Researchers have also tried using perturbation of learned policy for exploration (Fortunato et al., 2017; Plappert et al., 2017) and using value function estimates (Osband et al., 2016). Again these approaches have mostly been considered in context of external rewards and hence turn out to be sample inefficient.

## 7 DISCUSSION

Exploration has always been a crucial topic in robotics yet all the recent advances have been shown in either video-games or simulation due to sample inefficiency. This paper focuses on policy optimization for exploration policy learned via intrinsic rewards. Specifically, we propose a simple yet effective local approximation which allows us to perform policy optimization using likelihood maximization. We demonstrate the effectiveness of our approach on Sawyer robot which learns how to interact with objects (trained from scratch).

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

## APPENDIX A    TRAINING USING EXTERNAL REWARD

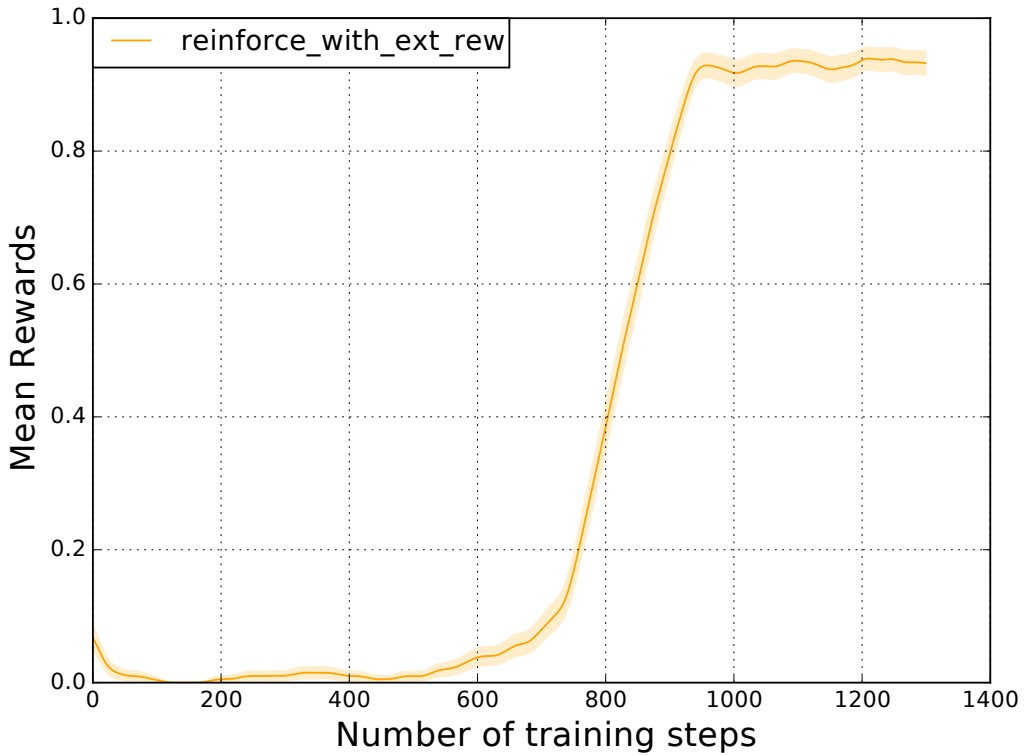

Figure 4: Used external reward from environment to learn policy

In this section, we tried to analyze how policy performs if it is provided with external reward from the environment. The environment gives +1 to the agent if it moves the objects, otherwise 0.

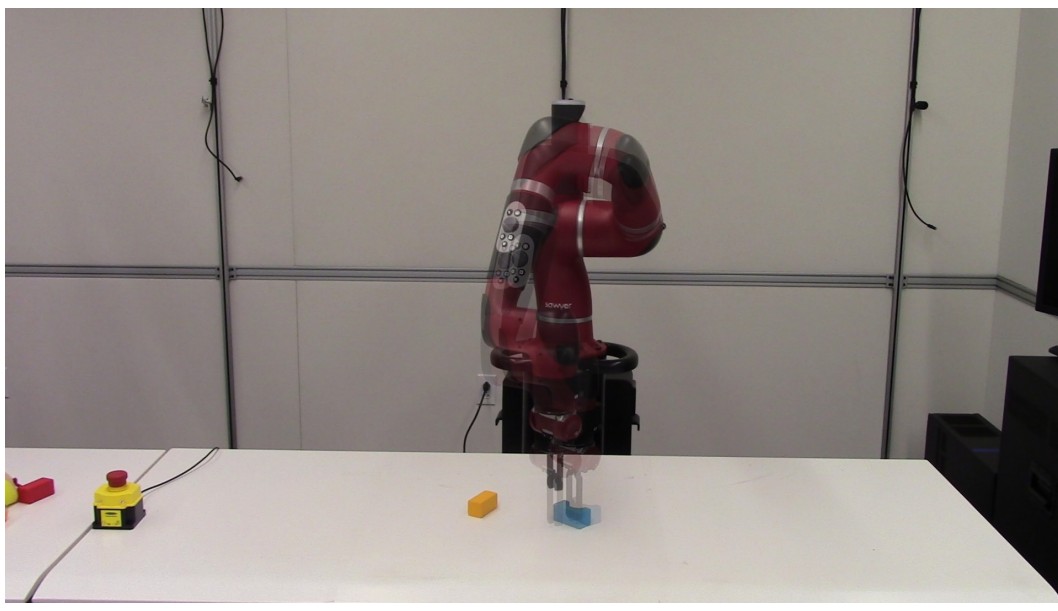

Figure 5: Robot Interacting with objects based on curiosity

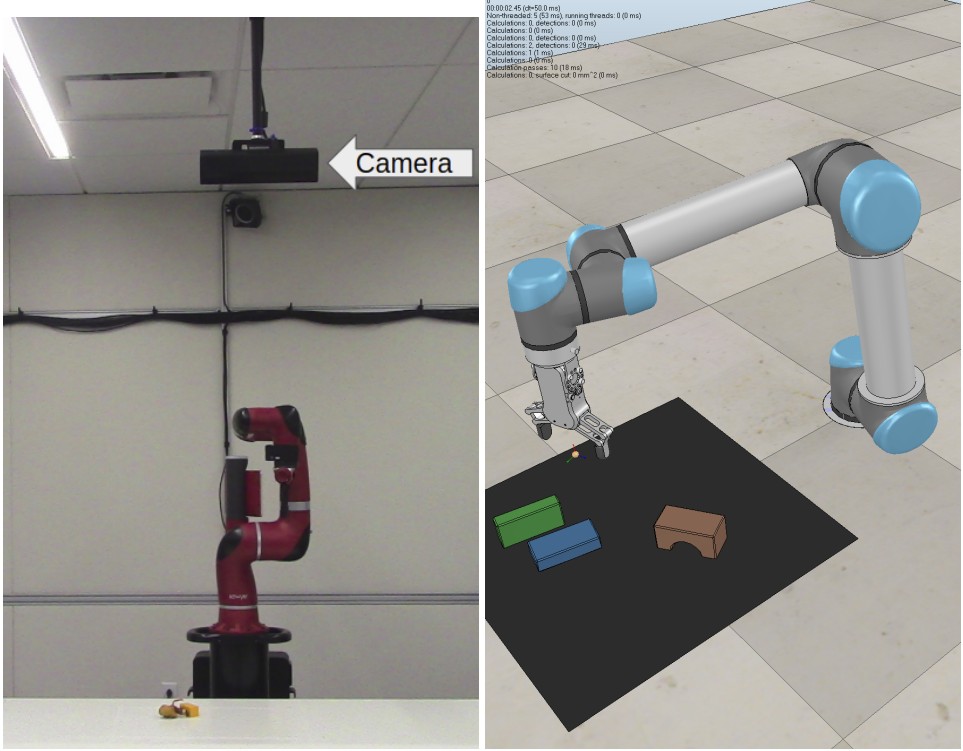

(a) real world robot setting                    (b) simulation setting

## APPENDIX B    PREDICTION LOSS

Prediction Loss We compare the prediction loss of the forward model learned on the data collected by different exploration schemes. We use L2 distance in the ImageNet feature space. As the result in the Table 1 indicate the data collected using our model performs better.

| Method Name | Free-Space | Pushing | Failed Grasp Attempt | Grasping | Mean |
|---|---|---|---|---|---|
| Exploration w/ REINFORCE | 0.28 | 0.23 | 0.17 | 0.64 | 0.33 |
| Exploration w/ Ours | 0.17 | 0.23 | 0.17 | 0.63 | 0.3 |

Table 1: **Prediction Loss** of the forward prediction learned on the data collected by the exploration policy for multi-object scenarios on real robot.

## APPENDIX C    COMPARISON ON ATARI GAMES

Figure 7: Comparison to recent state-of-the-art exploration strategy (Burda et al., 2018) on Atari.

We now demonstrate the comparison of our proposed exploration formulation in Section 3.2 with the recently proposed (concurrent work) large-scale-curiosity (Burda et al., 2018) implementation on benchmark Atari games. For baseline, we used the publicly available implementation off-the-shelf (Burda et al., 2018). The policy training is performed using the curiosity-reward function for the baseline and using our differentiable exploration objective for our method, without relying on any external rewards from the game. We measure the external reward from the game as a proxy to evaluate the quality of exploration. This external game reward is only shown for evaluation purposes but not used for training.

Our proposed model in Equation 3 relies on the learned prediction model to train the policy in an efficient manner. However, in the absence of any external rewards on these games, the exploration would have to rely on long-term prediction of the learned forward model to make progress that correlated with getting extrinsic reward. However, the long-term model learning is still an active area of research (Ebert et al., 2017). Since using current approaches, the long-term prediction models are hard to learn, so we use our objective in conjunction with reinforce objective when the long-term prediction quality matters. The orange curve in Figure 7 denotes the recent implementation curiosity baseline provided in the paper. Note both these environments require long-term modeling of the environment and on both the environments, our objective seems to provide gain in efficiency.

## APPENDIX D    ANALYZING THE DIFFERENTIABLE EXPLORATION OBJECTIVE

In this section, we discuss a toy experiment to investigate how would the proposed object behave due to the effect of local constancy approximation, explained in Section 3.2.

**Example-1: Investigating behavior of the objective**
Lets consider a simple setting of navigation in the environment where states are represented by the color of rooms that agent visits. We compare the behavior of count-based and curiosity-driven exploration with our proposed differentiable exploration objective. In both cases, let us assume a tabular representation (i.e. learns in one example) for the exploration model.

For the count-based exploration reward, count based method will only consider next state being visited irrespective of what path being take to reach over there. It will give $+1$ exploration reward for the first 2 transitions mentioned in Table 2. However, when agent does transition from green to blue, count based method will give 0 reward, even though this state transition happened for the first time. In contrast to this, a prediction-based curiosity objective gives $+1$ reward for the first three transitions because they all happened for the first time, and 0 reward for the last case.

| Timestep $(t)$ | State $(x_t)$ | Action $(a_t)$ | Next-State $(x_{t+1})$ | Count-Based Reward Value | Curiosity Reward Value | Our Action Gradient |
|---|---|---|---|---|---|---|
| 1 | white | $a_1$ | blue | 1 | 1 | $+ve$ |
| 2 | blue | $a_2$ | green | 1 | 1 | $+ve$ |
| 3 | green | $a_3$ | blue | 0 | 1 | $+ve$* |
| 4 | green | $a_3$ | blue | 0 | 0 | $-ve$* |

Table 2: Discussing the behavior of exploration reward for count-based method, curiosity (dynamics-based) and our differentiable exploration. $+ve$ and $-ve$ refers to the gradient direction for the action in our case. *,* refers to the cases when our objective of differentiable exploration behaves differently than count-based and curiosity.

Both curiosity and count-based objectives are optimized via policy gradients in general. Hence, when reward is 1 it means the action is encouraged and hence the probability of that action is increased as desired. However, the more interesting scenario is when the reward is 0. In this case, the policy gradient objective suppresses the current action being taken due to the reinforce objective (Schulman et al., 2015). Since the action probability is normalized across action, this amounts to increasing other actions in proportion to their existing log probability. This is fine in expectation, but suffers from high variance in small batches.

Now let us consider our proposed objective from Equation 3. In our case, the behavior in first three cases is similar to the prediction-based curiosity objective. However, in the final case, when the same transition is being executed, our model would explicitly try to force the policy to take action that does not lead to the *blue* as next state. Now if the environment contained mostly blue rooms, **our objective will explicitly encourage actions that go to a non-blue rooms, in contrast with reinforce which penalizes actions that lead to blue rooms.**. We now elaborate why this simple difference in behavior has a striking impact in high dimensional scenarios.

**Example-2: Investigating gradient direction**
Consider a thought experiment of a white surface with a red magic box kept at a random location. The action space of a robotic arm involves going to any location (e.g. a discrete $n \times n$ grid) on the surface.

If the robotic arm end-effector ends up in the area occupied by magic box, something unpredictable happens. Now since the area of red magic box is much smaller compared to the white surface, it would take long time to hit the magic area randomly. Hence, most of the example for learning dynamics will be from the free surface. Suppose a forward dynamics model is learned on a bunch of initial sample interactions (out of which most of them will be on free space) and hence the models prediction error will be very low in white surface and high the red magic area. A curiosity-objective learned with *reinforce would penalize the actions leading to white space* as the reward will be low over there, and the probability of other actions (including going to red area) will increase in proportion

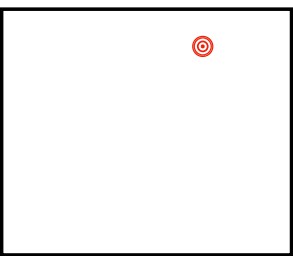

of their log-probs. In contrast, *the direct gradients in our case would directly force the actions to go to non-white locations* and hence, quickly reach the locations in red area. This examples is representative of scenarios where the number of un-interesting interactions is much more than the interesting interactions which are worth exploring, which is usually the case in the real-world setups.

### D.1 THEORETICAL UNDERSTANDING OF OBJECTIVE

The proposed objective in Equation 3 bears similarities to the experiment-design in optimization literature. For a detailed overview, refer to the Chapter-7 in Boyd & Vandenberghe (2004). The goal of experiment design is to optimize for the actions such that the error covariance in the model is small, which means going to areas with high prediction error. While this is out of the scope of paper, we hope that this connection would help spur discussions for future work to build upon.

