# OpenReview forum: "Beyond Games: Bringing Exploration to Robots in Real-world"
_ICLR.cc/2019/Conference_

### Official Review · AnonReviewer2 · 2018-11-02

**Rating:** 5
**Confidence:** 3

**Review:**

This paper presents an interesting way to reformulate intrinsic curiosity as a differentiable function. The authors compare the differentiable function against using prediction error via REINFORCE and DQN, showing that their intrinsic curiosity method results in more interactions with unseen objects than the other two methods. For DQN this is to be expected, but it shows that backprop through this function is more efficient than reinforce in getting to unseen state spaces. I think this is an interesting method/proposal and is a somewhat novel reformulation of intrinsic error, but I do have some concerns in comparisons/claims.

In the introduction, the authors say that the intrinsic curiosity method proposed by Pathak et al. is sample inefficient and isn’t tested in robots. However, to my understanding the REINFORCE baseline isn’t really equivalent (though it may be possible that it is, it was unclear how exactly the loss was formulated in the baseline, did include the other components from Pathak et al.?). If the claim is that this method is more efficient, I think it should have compared against that method directly.

Moreover, I think the description of the experiments doesn’t provide enough information. For example, the method says that different learning rates were used for the min-max game to stabilize it, but doesn’t say what they were.
Also, for the DQN baseline what were the parameters? Was there an epsilon greedy policy on top of the exploration reward? Was this annealed as in other work? Generally, I think more detail is needed throughout (even if it just refers to a more detailed appendix).

Overall, I think this work needs to be revised to include more details on hyperaparameters, details on the baselines, and describing differences between Pathak et al.’s method and the REINFORCE baseline. Moreover, feedback from other comments on this work should be addressed which reflect in more detail my comments below on opinionated claims (e.g., https://openreview.net/forum?id=SkzeJ3A9F7&noteId=HJlFlZOa2X )


Comments/Thoughts:

+ I think in the introduction there are some statements that probably need citations. For example, “But the same formulation from an optimization viewpoint, it suffers from all the bad properties of extrinsic rewards. The reward is a function of environment behavior with respect to the performed action. Since the environment behavior function is unknown, it is treated as black-box and hence the gradients have to be computed using REINFORCE (Williams, 1992) which is quite sample inefficient.” —> Why is this true? Is there a citation that can back this? Do you prove it later in the paper?
+ “Yes, 54 environments but no real-world physical robots” —> this and the intro seems like a blogpost at times. That can be fine (some would argue it’s a good thing), but there seem to be some opinions without citations/backing, I suggest trying to back up statements wherever possible and avoid opinions. For example in this statement, robots aren't a requirement for evaluating intrinsic motivation.
+ “Since the environment behavior function is unknown, it is treated as black-box and hence the gradients have to be computed using REINFORCE (Williams, 1992) which is quite sample inefficient.” —> citation/backing? it might be nice to point to the experiment section here to back it (e.g., "As will be shown in Section X and in \citet{something}, REINFORCE can be quite sample inefficient")
+ “In practice, the existing on-policy algorithms, e.g., A3C (Mnih et al., 2016), PPO (Schulman et al., 2017) etc. are deployed off-the shelf -> This is confusing, so is this using REINFORCE or PPO/A3C? what is this statement referring to?
+ “regress to rti to learn value estimates (i.e., off-policy) as discussed in the previous section” —> regress to \sum r_t{I} for a value estimate?? Value is the expected return so not sure if this is a typo or i missed something earlier
+ What is the actual loss function used for the baseline? Is it the same as Pathak et al.?
+ What are the hyper parameters for DQN exploration? What are all the hyper parameters for any/all the algorithms?
+ Was a variance-reducing baseline used in REINFORCE?
+ What is the variance representing in the graphs, std across several trials? Maybe I missed it, but how many trials represent this standard deviation?
+ “Hence, we train the forward predictor slightly faster than the policy by keeping higher learning rate to stabilize the learning process. “ —> what were the learning rates?


Linguistic/Typos:

Also, some minor, but frequent, grammatical issues/typos that I’ve added below could be fixed. I would ask that the authors please have the submission proof-read for English style and grammar issues. There are many minor mistakes, some of which I’ve tried to point out below.

+ “This leads to a significantly sample efficient exploration policy. “ —> significantly more (?) sample efficient ?

“Why is that? To understand the reason behind sample inefficiency of curiosity or intrinsic rewards, notice how the intrinsic rewards are given by agent” —> by the agent?

“Forward model fθF is trained to minimize its loss which amounts to minimizing rti with respect to θF” —> the forward model

“However, policy is optimized to maximize the objective” —> However, the policy

“We can also optimize  for policy parameters θP via differentiable loss function” —> We can also optimize for (the) policy parameters \theta via (a) differentiable loss function?

“To optimize policy to maximize a discounted sum “ —> To optimize the policy

“How good is Forward Prediction Model” —> How good is the forward prediction model

There are several other spots, but basically another pass over the paper might be worth it to check for these sorts of issues.

---

> ### Author Response · Authors · 2018-11-20
> **[Authors' Response to R2] Hyper-parameters and other details of experimental setup**
>
> We would like to first thank the reviewer for the comments. Please see the meta response in the combined reply for the common concerns among reviewers. The reviewer found the formulation to be "interesting" and "novel" but has raised concerns about lack of details about the experiment hyper-parameters. We answer them below:
>
> [R2]: REINFORCE baseline vs. Pathak et. al.? Is it the same as Pathak et al.?
> => Please refer to the combined response to all reviewers above. It is indeed the same.
>
> [R2]: "different learning rates .. but doesn't say what they were"
> [R2]: "What were DQN parameters? Epsilon greedy"
> [R2]: "include more details on hyper-parameters ..."
> => Yes, in our case, we can simply train the forward model faster than policy to stabilize the training. The learning rate for the model was 5e-4 and policy 1e-4 with an entropy loss coefficient of 1e-4. The optimizer used was Adam.
>
> Yes, we used standard DQN algorithm with prioritized experience replay [Schaul et. al. 2016] and epsilon-greedy to maximize the curiosity objective [Pathak et.al. 2017]. Q-value was trained with a learning rate of 1e-4. In epsilon-greedy, epsilon was initialized at 0.5 then annealed over training to 0.1.
>
> We agree that these details are important for the reproducibility of work, and we will include them in supplementary -- we apologize that we missed adding them in the supplementary. That being said, we will make our code, environment, and models public upon release.
>
> [R2]: "Feedback from the other commentator should be addressed"
> => We have answered both the factual and stylistic comments with proper citations in our reply.
>
> [R2]: "... This is confusing, so is this using REINFORCE or PPO/A3C? .."
> [R2]: "it might be nice to point to the experiment section here to back"
> => With due respect, we believe reviewer has misunderstood the background here. REINFORCE is an operator that was proposed in (Williams, 1992), and then became the main ingredient of policy gradient algorithms. All policy gradient algorithms, whether PPO or A3C, use reinforce operator to update the policy. PPO, in addition to reward maximization, penalizes KL-divergence change while A3C relies on regularization from multiple workers, but both these algorithms use reinforce at the core of it.
>
> [Schulman et.al. 2016] provides a nice discussion on how REINFORCE helps to optimize any black-box function in expectation using only samples from the function. More detailed and fundamentals of policy gradients are described in [Sutton & Barto, 1998]. We implemented the policy-gradient algorithm as optimizing the curiosity objective in [Pathak et.al. 2017] as a baseline which we call "REINFORCE". We will update the legend of graphs to say [Pathak et.al. 2017] in future to avoid confusion.
>
> [R2]: "regress to rti .. discussed in the previous section .. so not sure if this is a typo"
> => Thanks, it should be indeed summed. However, note that in the previous section it was mentioned correctly. We will correct this typo.
>
> [R2]: "Some minor, but frequent, grammatical issues/typos .."
> => Thank you. We will fix all these writing typos and make sure there no other ones.

---

> > ### Comment · AnonReviewer2 · 2018-11-21
> > **Reply to specific comments**
> >
> > "With due respect, we believe reviewer has misunderstood the background here. REINFORCE is an operator that was proposed in (Williams, 1992), and then became the main ingredient of policy gradient algorithms. All policy gradient algorithms, whether PPO or A3C, use reinforce operator to update the policy. PPO, in addition to reward maximization, penalizes KL-divergence change while A3C relies on regularization from multiple workers, but both these algorithms use reinforce at the core of it."
> > --> Sorry if I was not clear here. The difference between REINFORCE and PPO is self-evident, and that was not my point. I was trying to suggest that the language highlighted made it unclear as to why PPO is mentioned if the REINFORCE baseline is under test (e.g., the language suggests that some aspect of PPO/A3C is relevant here, but that doesn't seem to be the case since PPO nor A3C are compared against).
> >
> > "We agree that these details are important for the reproducibility of work, and we will include them in supplementary -- we apologize that we missed adding them in the supplementary. That being said, we will make our code, environment, and models public upon release." --> I appreciate the addition of these important details and look forward to seeing them in revisions.
> >
> > While the new revision is significantly more refined, I still don't think there is enough information about the experiments provided in the main text to judge the contributions properly in their current form (for example much of the information that was asked about above, whether REINFORCE in this paper uses a variance reduction baseline, learning rates, etc. still remain missing). Moreover, I agree with the concerns that other reviewers raised regarding the baselines. For example, while it is understandable that the authors compare against optimizing this curiosity reward through REINFORCE vs. their method, it is unclear whether the REINFORCE method used here makes use of variance reduction methods which are known to boost performance significantly. However, I have updated my original review rating because I believe this current draft is much more clear than the prior version.

---

> > > ### Author Response · Authors · 2018-11-25
> > > **Authors' Follow-up Response to R2**
> > >
> > > We thank you for your earlier feedback and are grateful for your follow-up response. We would like to clarify the subtle misunderstanding here. To keep the changes minimal for the reviewers in our updated draft, we only included new experiments and analysis in the last two pages of the supplementary in the paper, which were not possible to include within the text of rebuttal responses. We are yet to include the information and details provided as part of our rebuttal responses in the main paper. For instance, we described the learning rate, optimizer, epsilon-greedy, experience replay details etc. in our first response to your main review (above) and will include them in the final draft of the paper, posting here again just for reference:
> > > "The learning rate for the model was 5e-4 and policy 1e-4 with an entropy loss coefficient of 1e-4. The optimizer used was Adam. We used standard DQN algorithm with prioritized experience replay [Schaul et. al. 2016] and epsilon-greedy to maximize the curiosity objective [Pathak et.al. 2017]. Q-value was trained with a learning rate of 1e-4. In epsilon-greedy, epsilon was initialized at 0.5 then annealed over training to 0.1."
> > >
> > >
> > > Variance Reduction: Yes. Since our robotic pushing environment is not long term, the value-function baseline reduces to a simple constant-value in our scenario, which we used for variance reduction. It turned out to be sufficient to make the REINFORCE algorithm work efficiently in this environment using external rewards. We will add this to the paper. Please refer to the result in the Appendix Figure-4, which shows that our REINFORCE baseline indeed works well with an external reward.
> > > In order to further demonstrate that our baselines are credible, we took the public implementation from the recent large-scale curiosity paper [Burda et. al. 2018]. We use the curiosity baseline provided in the paper and to demonstrate our objective function for exploration is meaningful, we add our objective to the original exploration objective. This comparison uses the official implementation of PPO with all the necessary state-of-the-art techniques for variance reduction. Please see results in Appendix Section-C of the updated paper.
> > >
> > > Thank you again for the review. We look forward to the follow-up comments.

---

### Official Review · AnonReviewer3 · 2018-11-06
**Review: Clear reject**

**Rating:** 3
**Confidence:** 4

**Review:**

Summary:

This paper proposes a novel differentiable approximation to the curiosity reward by Pathak et al. that allows a learning agent to optimize a policy for greedy exploration directly by supervised learning, rather than RL.
The authors motivate this work with arguments about the sample-efficiency required by real robot learning, and demonstrate basic results using a real robot.

Comments:

The paper has serious style and tone issues that must be addressed before publication. The rest of my review will focus solely on the technical details.

The experimental details are lacking (learning rates? rollout lengths for REINFORCE? what are the inner and outer loops and what are their sizes? what are the plots measuring - extrinsic reward? intrinsic reward? what is "multi-step learning" in Table 2?).
Without these details, the results will be difficult to validate and reproduce independently.

The approach is compared only against very weak baselines. Why vanilla REINFORCE and not any of the modern policy-gradient algorithms (A3C, PPO, TRPO, DDPG, ...)? The ability to deal with this large action space is certainly impressive, but it is likely far too large for DQN or REINFORCE to work, so the comparison is questionable to begin with: you can make any algorithm fail if you give it an unnecessarily difficult interface. Did the authors try a smaller, more traditional action space?

The authors claim the sample-efficiency improvements by many existing exploration approaches are insignificant. In what way are the results in this paper more significant? Table 1 shows very minor improvements to a MPC planning task, Appendix Figure 1 shows barely any improvement over the baseline, and Appendix Figure 4 shows that learning from extrinsic rewards using REINFORCE seems to work just fine. Why use intrinsic rewards at all in this case? It appears that maybe some of the results look significant because the baselines are so weak.

The paper contains many factual errors and unsupported claims. For example:
- "the field of RL was born out of need to make our robots learn"
- "none of the recent advances have translated to success in the field of robotics" (see e.g. the proceedings of CoRL 2017 and 2018)
- "Building a good model will require enormous number of interactions" (see e.g. PILCO)
- "[our approach enables us to] for the first time ever, implement exploration on a real-world physical robot" (PILCO and many others)
- describing Pathak et al. curiosity as a "gaussian density model" in eq1; it's a deterministic forward model
- in sec3, "regress r^i_t to learn value estimates", this is probably meant to be the discounted sum of rewards
- also sec3, "[REINFORCE] gives no signal as to what action to take"; the signal has high variance but it works (see all policy gradient work)

These errors can be easily corrected. However, the contribution of the paper is based on a more serious error:
- sec3.1, "If the policy could be optimized using direct gradients, the rewarder could ... inform the agent to change its action space in the direction where forward prediction loss is high."
This is incorrect. The paper is based on using the gradient of the forward model to directly optimize the policy to produce higher prediction errors, as in Pathak et al.
But in order to make the prediction error differentiable, it makes the severe assumption that the next state x_{t+1} is constant and does not depend on a_t, which is false and invalidates the idea of optimizing actions for prediction error.
As a result, the gradient obtained does not actually move the policy toward higher prediction errors.

To understand what the author's approximation actually does, consider a perfect forward model. No matter what actions the policy produces, the prediction error is always zero, but the authors' gradient is not. So it can't be optimizing for higher prediction errors.
Instead of optimizing for high prediction errors as the authors claim, the policy is being optimized for state transitions that are maximally different from the observed state x_{t+1}.

This is an interesting objective to optimize. I can see how it could result in interesting exploration. But it's not what the authors say they're proposing.
It's much more like a count-based exploration strategy, which prefers visiting states that are maximally different from the states visited so far. It is much less like the prediction-error based curiosity of Pathak et al. that the authors are motivated by.
I would like to see focused analysis of this particular objective. For example, would this not result in the policy oscillating between different parts of the state space, since it's only optimizing for maximal difference to what it just saw, rather than long-term knowledge gain? This issue requires more discussion.

Finally, the approach is not really robot-specific despite the title and arguments in the paper. I recommend pursuing a more general investigation, because if this objective is truly as effective as the authors believe, then it should be applicable in a wide variety of domains (many of which are very easy to evaluate in: Atari, OpenAI Gym, DMLab, VizDoom, Mujoco, etc.).

Conclusion:

The paper proposes an interesting new objective, but it is motivated by a very naive approximation that completely changes the behavior of the exploration compared to what the authors want to approximate. The idea is novel and worth exploring, but the paper should be heavily rewritten to emphasize what the authors are actually doing with this new objective, and should include thorough analysis of its behavior, before I can recommend acceptance.

---

> ### Author Response · Authors · 2018-11-20
> **[1/2][Authors' Response to R3] Interpretation of the objective under the local approximation**
>
> We would like to first thank the reviewer for the comments. These comments
>
> [R3]: "The experimental details are lacking"
> => We agree that these details are important for the reproducibility of work, and we will include them in supplementary -- we apologize that we missed adding them in the supplementary. That being said, we will make our code, environment, and models public upon release.
>
> The learning rate for the model was 5e-4 and policy 1e-4 with an entropy loss coefficient of 1e-4. The optimizer used was Adam. We used standard DQN algorithm with prioritized experience replay [Schaul et. al. 2016] and epsilon-greedy to maximize the curiosity objective [Pathak et.al. 2017]. Q-value was trained with a learning rate of 1e-4. In epsilon-greedy, epsilon was initialized at 0.5 then annealed over training to 0.1.
>
> [R3]: What are the plots measuring - extrinsic reward? intrinsic reward?
> => The plots are measuring the how frequently the robotic arm touches the object which is a proxy for measuring how good the exploration is. Note that this reward was not used for training the policy, but just used to measure the quality of exploration at training. The robot was trained with an intrinsic reward only.
>
> [R3]: "The approach is compared only against very weak baselines."
> [R3]: "I recommend pursuing a more general investigation ..to evaluate in: Atari"
> => Inspired by reviewer's suggestion, we have added a comparison on games in the supplementary using the state-of-the-art implementation proposed couple of months ago in (Burda et. al. 2018) [2]. Please see the combined response above for more details.
>
> Our paper is just a first step towards explicitly using the structure of the prediction model in training the policy in a supervised manner. It relies on the model being good which is true for short-horizon tasks, but learning good long-term models is still an active research area [Finn et.al., 2017, Ebert et.al. 2017]. Hence, our approach might provide diminishing gains for long-horizon tasks.
>
> [R3]: "this large action space is certainly impressive, but it is likely far too large for DQN or REINFORCE"
> => Our fundamental goal is to test exploration algorithms on real robotics setup where it is crucial to experiment in presence of large input space as well as large action space. We believe working with larger action space should be seen as a positive feature, and not something made to fail baselines. However, upon the reviewer's suggestion, we have added a comparison on games in the supplementary. Please see the combined response above for more details.

---

> > ### Author Response · Authors · 2018-11-20
> > **[2/2][Authors' Response to R3] Interpretation of the objective under the local approximation**
> >
> > [R3]: "Table 1 shows very minor improvements to a MPC planning task"
> > => As discussed in our combined response above, the baselines compared to in paper are same as the curiosity-driven exploration and our method achieves absolute 10% improvement in the mean accuracy for multi-object manipulation -- which we believe is a significant improvement.
> >
> > [R3]: “Appendix Figure 4 shows that learning from extrinsic rewards using REINFORCE... the results look significant because the baselines are so weak."
> > => Please note that the external reward curve shown in Figure-4 is just to show that the REINFORCE baseline works in our setting. The reward, in that case, was touching the objects, while our single exploration policy learns diverse behavior without rewards learns to push, pull, grasp objects allowing us to collect useful data for training the planning models. A more detailed discussion on the usage of extrinsic and intrinsic rewards is presented in (Burda et. al. 2018) [2].
> > => We have now added comparisons to state-of-the-art off-the-shelf baselines on Atari games.
> >
> > [R3]: "Building a good model require enormous number .." (see e.g. PILCO).
> > [R3]: " .. implement exploration on physical robot" (PILCO and many others)
> > => These are great advances and strong motivation for our work. We already discussed them in our method section (see Section-3.1, first para). Note that these are not reinforcement learning methods and based classical optimal control using learned models. However, those models are dynamics models in low-state-space of the agent learned using Gaussian processes and not from high dimensional raw image input. Even using deep networks, these methods have trouble generalizing to raw images as input (e.g. Deep PILCO [Gal et.al. 2016]) and rely on the learned model is quite good. Moreover, PILCO is not an exploration approach but learned for an end-task with well-defined cost objective.
> >
> > [R3]: "To understand what the author's approximation actually does, consider a perfect forward model … is always zero.. maximally different from the observed state"
> > => If the prediction error is always zero, the action gradients will also be 0. The interesting scenario happens, when the error is low but not zero, in which objective encourages action that leads to states different from x_{t+1}. We have added a detailed clarification in Appendix Section-D.
> >
> > Further, we re-emphasize that our local constancy approximation for x_{t+1} is only applied in a very local neighborhood of the executed action. It is only used for computing *local* gradients and nowhere else. This means when a completely different action a_t_new is applied at x_t, we will consider the new next state corresponding to a_t_new for computing gradients at a_t_new. One interesting consequence of our objective is that it has to be trained in an on-policy manner as off-policy violates the local-constancy approximation.
> >
> > [R3]: "It's much more like a count-based exploration … much less like the prediction-error based"
> > => We agree with the reviewer that our is not exactly the same as Pathak et.al., and we did not claim this as well. However, we respectfully disagree that this objective is more-like a count-based strategy. This is so because our constant state approximation is only used in the small neighborhood of the executed action.
> >
> > Thank you for bringing this point up. Inspired by this, we have added a detailed discussion in Appendix Section-D on understanding the behavior of objective.
> >
> > [R3]: The paper has serious style and tone issues that must be addressed before publication.
> > => We will update and soften the introduction of the paper as discussed in the combined response above.

---

> > > ### Comment · AnonReviewer3 · 2018-12-07
> > > **Reviewer response to clarifications**
> > >
> > > Thank you for the clarification of the experimental details, and for the extra discussion of the new objective.
> > >
> > > Unfortunately I'm still skeptical about the precise behavior of the objective: if you're always pushing the policy toward an outcome that's as different as possible than the previous outcome, does this result in non-convergent oscillation in policy space? In Pathak et. al the predictive model should eventually converge, so the intrinsic rewards should eventually go to zero once exploration is complete. For this proposed objective, I don't see a reason it should converge because there's always a direction to push the policy toward different outcomes, and the process has no memory mechanism to avoid repeating previous exploration. This requires investigation, or at least an explanation of why this is not a problem.
> > >
> > > Once again, I think it could be an interesting objective to study, but I'd like to see a version of this paper that is more focused on analysis of this objective as an exploration strategy and less about the specific application of robotics, which seems unrelated to the exploration angle. I appreciate the experiments on the Atari games, but it looks like the improvements over the baseline are minor (with high variance) and the BeamRider results in the Burda et. al paper achieve roughly twice the score that you report from your baseline, which makes me wonder whether the performance reported here is representative.
> > >
> > > As before I think there is some potential here; the idea of turning an exploration procedure from RL into a supervised task is interesting. At this point I still have concerns about convergence and oscillation, so my decision remains.

---

> > > > ### Author Response · Authors · 2018-12-10
> > > > **Follow-up Response to R3's comments**
> > > >
> > > > Thank you for the reply. We clarify the concerns here:
> > > >
> > > > R3: "the BeamRider results in the Burda et al. paper achieve roughly twice the score that you report from your baseline, which makes me wonder whether the performance reported here is representative."
> > > > => Curiosity results based on random features are the most stable and best-performing method in Burda et.al. on Atari. Hence, we used random features for both our method as well as the Burda et.al. in Figure 7. The results indeed *exactly* matches the curve in Burda et.al. Please compare: orange curve in Figure 7 of our paper vs. green curve in Figure 2 of Burda et.al. -- they have the same performance.
> > > >
> > > > => We think the reviewer has mistakenly compared VAE to random-feature based curiosity results. It turns out the VAE features worked the better only in the BeamRider game and the simplest/best-performing method in Burda et al. is based on random-features which we used. We will try our method on top of VAE features as well.
> > > >
> > > >
> > > > R3: "If you're always pushing the policy toward an outcome that's as different as possible than the previous outcome"
> > > > => We would like to emphasize that our objective does not *exactly* push the policy toward an outcome that's different. This is so because our constancy assumption only applies to compute gradient in only a very local region around the exected action. The reviewer would be correct if we were to make this assumption at all points, but we only use it while computing gradient only at the point when action is executed (i.e., pi(s_t)=a_t and a_t is applied to obtain the corresponding s_{t+1}) in an on-policy manner.
> > > >
> > > > => When applied locally, this assumption results into an interesting behavior which behaves like count-based *locally*, and prediction-based *globally*. This results into moving the agent's policy towards the actions which would result in a different outcome locally (like count-based because s_{t+1} is fixed) and towards the actions where prediction model is wrong globally (because we also update s_{t+1} using the new transitions obtained by updated agent's policy). Please see our example-2 in section-D for an illustration of how this idea especially plays out well in robotic manipulation settings.

---

> > > > > ### Comment · AnonReviewer3 · 2018-12-10
> > > > > **Response**
> > > > >
> > > > > "We think the reviewer has mistakenly compared VAE to random-feature based curiosity results"
> > > > >
> > > > > My mistake. Thank you for the clarification on this, I was indeed looking at the best curve, rather than the random features. However, given that the VAE features performed best on this game, I think it would be reasonable to compare to those instead. Or if you would like to compare to the random features, I think it would be fair to compare on another game where the random features do well (for example, what about Breakout, Pong, Montezuma, Mario, Qbert, Seaquest, Space Invaders, do you have results for those?).
> > > > >
> > > > >
> > > > > "We would like to emphasize that our objective does not *exactly* push the policy toward an outcome that's different... When applied locally, this assumption results into an interesting behavior which behaves like count-based *locally*, and prediction-based *globally*."
> > > > >
> > > > > I don't understand the explanation for why this is the case. The gradient used to update the policy does exactly push in the direction that maximizes the difference in outcome. In what way does taking a difference-maximizing step imply a global trend toward maximizing prediction error?
> > > > >
> > > > > I don't see the necessary connection between achieving Euclidean-distant outcomes and achieving outcomes that are hard to predict; I think that would need to be established rigorously in order to make the claim that's being made here.

---

> > > > > > ### Author Response · Authors · 2018-12-10
> > > > > > **Mathematical description of the local assumption**
> > > > > >
> > > > > > Thank for the follow-up comments.
> > > > > >
> > > > > > R3: "I don't see the necessary connection between achieving Euclidean-distant outcomes and achieving outcomes that are hard to predict; I think that would need to be established rigorously"
> > > > > > => We describe our intrinsic reward r_t mathematically as follows:
> > > > > >
> > > > > > r_t = || f(x_t, \pi(x_t)) - y_t ||
> > > > > >
> > > > > > where
> > > > > >     if \pi(x_t) == a_t:
> > > > > >           y_t is set to x_{t+1}, i.e., T(x_t, a_t)  -- kept constant around a_t (local)
> > > > > >     else \pi(x_t) != a_t:
> > > > > >           y_t = T(x_t, \pi(x_t))             -- updated to the true next state (global)
> > > > > >
> > > > > >     - f is learned forward model
> > > > > >     - T is true environment dynamics (unknown)
> > > > > >     - \pi is the agent's policy
> > > > > >     - {x_t, a_t, x_t+1}: a real transition executed in the environment
> > > > > >
> > > > > > => Here, only the first step of gradient enforces euclidean-distant outcome to y_t (i.e., count-based 'locally'). But as soon as the first gradient step is taken, the policy gets updated, and hence, \pi(x_t) != a_t. Therefore, the target y_t will also get updated to the new ground truth next state, as shown in the else condition above.
> > > > > > => Hence, next gradient step will not try to maximize the difference from original y_t but with the new next state obtained by T(x_t, \pi(x_t)). This implies that, in order to maximize the objective r_t in the global sense, the agent policy should take actions where f(x_t, a_t) will be far from *corresponding* ground truth x_t+1 -- which is same as prediction error being high.
> > > > > > => We discuss the potential benefit of this simple *local* trick in Example-2 of Section-D in the appendix.
> > > > > >
> > > > > > We again emphasize that we would rework the introduction and have shown results in simulation as well as real robot exploration in a high-dimensional input (images) and high-dimensional action space setup. Hopefully, the reviewer can take these points into consideration.

---

### Official Review · AnonReviewer1 · 2018-11-07
**Laudable goal, but paper not good enough**

**Rating:** 3
**Confidence:** 5

**Review:**

Beyond Games: Bringing Exploration to Robots in Real-world
===========================================================

This paper tackles the laudable goal of making an algorithm for efficient exploration in "real-world" RL.
To do this, they augment the "curiosity" algorithm of Pathak et al with a differentiable approximation to the reward prediction model.
They motivate this algorithm through several intuitive arguments together with a series of experiments where the algorithm outperforms vanilla DQN/REINFORCE.


There are several things to like about this paper:

- The problem of making "real-world" practical algorithms for exploration is clearly one of the biggest outstanding problems in reinforcement learning.

- The authors have sucessfully gone from ideas, to algorithm, to real robot and their algorithm really seems to outperform the baselines.

- The authors clearly make an effort to survey a wide variety of recent papers in the field



However, there are several important places where this paper falls down:

- In a paper that posits a new, groundbreaking, real-world application of "exploration" there is remarkably little discussion of the key issues of "efficient exploration". Indeed, I don't think that this paper even presents a clear metric for how we can tell if something *is* a good method for exploration.
  + This is a huge shortcoming, since we know that it is possible to guarantee polynomial regret bounds for many settings (mostly tabular, but some with function approximation too... see UCRL2, PSRL and more)... there is no discussion of whether the proposed algorithm would also satisfy these bounds?
  + Of course, this is not a paper designed for "tabular MDPs", but we already have exploration algorithms like UCB / Thompson sampling that *are* widely used in online advertising... so why is this method not compared/contrasted to these approaches?

- There is very little *science* in this paper, beyond the experiments pitting "improved algorithm" vs DQN/REINFORCE, which nobody ever claimed would be a good approach to exploration! I don't think it's possible to assess if their algorithm (which I don't think has a clear name beyond "sample-efficient exploration formulation") performs better than the myriad of other exploration approaches listed. Although many intuitive arguments are presented, I did not find these convincing, and the overall narrative ends up being a little jumbled.

- A lot of the writing is generally imprecise, and alludes to claims/statements that make no sense to me:
  + "... most of these sucesses have been demonstrated in either video games or simulation environments. This is primarily becuase the rewards (even the intrinsic ones) are non-differentiable ..."
  + "Again these approaches have mostly been considered in context of external rewards and hence turn out to be sample inefficient"
I would suggest that each statement/claim is backed up by some material reasoning/statement/experiment unless extremely obvious - at the moment these are not!

- Nothing in this algorithm really seems specific to real-world... or at least nothing in the competing algorithms seems to preclude them from being run on a real-world robot... I think that the main issue is that if people want to iterate fast (or don't have a robot) they prefer to do things in simulation. If your point is really that findings from simulation don't translate to real robots, then I think that is really interesting, but I don't see any evidence for that in this paper.


Overall, it is clear that this is an interesting area to do work in.
The goal of making a practical algorithm for real-world exploration tasks is exciting.
However, in its current form, this paper falls well short of the level of science and insight I would expect for ICLR.

---

> ### Author Response · Authors · 2018-11-20
> **[Authors' Response to R1] DQN/REINFORCE include exploration reward; Comparison to UCB**
>
> We would like to first thank the reviewer for the comments. Please see the meta response in the combined reply for the common concerns among reviewers. We answer other individual concerns below:
>
> [R1]: Only comparison to "DQN/REINFORCE, which nobody ever claimed would be a good approach to exploration!"
> => We respectfully disagree with the reviewer as there seems to be some misunderstanding. Here, DQN/REINFORCE do not refer to the vanilla RL algorithm but are optimizing the curiosity exploration objective proposed in Pathak et.al. [2017]. Please see the combined response above for details. We hope that this answers the reviewer's major concern about the comparison to other exploration methods.
>
>
> [R1]: Comparison to UCB / Thompson sampling; UCRL2, PSRL.
> => As the reviewer agrees, most of these algorithms have been primarily studied in tabular cases and there have been some extensions to function approximators. These algorithms do not scale to the high dimensional image input (320x320x4) and a high dimensional action space (location, angle, push, grasp) used in our paper.
>
> Further, we respectfully believe there is a misunderstanding about the context in which the term "exploration" has been used in this paper. Our robotic agent explores the environment completely out of its own; using only its intrinsic reward and no external reward at all. UCB-like algorithms are parameter-space exploration and are successful in learning a policy in presence of external reward (even if sparse). One can not use UCB when there is no external reward as the value functions would collapse.
>
> The algorithms UCRL2/PSRL integrate the prior over dynamics or reward function into learning a policy in a natural and efficient manner. These have been great advancement but not relevant to the problem setup or the goal of this paper because our agent's start completely from scratch without any prior, and has access to no external rewards.
>
> As far as bounds are concerned, our algorithm draws a similarity to Experiment-Design literature from optimization. In a linear case and explicit policy, optimizing our objective is the same as finding a max-variance action which is an efficient way to discover the space satisfied by the model. While this is out of the scope of the paper, we have mentioned the pointers in Appendix Section-D.1 and elaborate more to spur discussion for future work to build upon.
>
>
> [R1]: "how we can tell if something *is* a good method for exploration".
> => It is difficult to measure exploration as an end in itself. There are several domain-specific metrics can be created that could act as a useful proxy to measure the quality of exploration. For instance, we measure how frequently the robot touches the object kept on the table purely out its exploration, and found out that this frequency increases to almost perfect as our training proceeds (please see the video on the website link provided). Another proxy to measure exploration is to see if a purely exploration-driven learning without using any external reward can help the agent obtain an external reward, as discussed in (Burda et. al. 2018) [2]. However, this latter proxy is only applicable to games-like environments.
>
> In this paper, we argue that the true measure of exploration is how well it is able to contribute to learning planning models later. Using the data collected by the exploration, we build models and report the success of those models for grasping, pushing etc. If the robot explored well during exploration, then it should have gathered good data for learning planning models for manipulation. This discussion is present in Section-5.1, 5.2 of paper. We will clarify it further.
>
>
> [R1]: "Nothing ... seems specific to real-world... or at least nothing in the competing algorithms seems to preclude them from being run on a real-world robot…"
> => The REINFORCE comparison is indeed existing curiosity-driven exploration from Pathak et.al. [2017] and being run on the real robot (please refer to meta combined response above for details)
>
> We further provide a comparison for long-term horizon dependency tasks in video games. Please refer to the combined response for details.

---

### Public Comment · (anonymous) · 2018-11-05
**Unsubstantiated claims, grandiose writing, poor experiments**

I appreciate the intention of the authors to try curiosity-based learning on real-world tasks like robotic manipulation. It is definitely an important problem to work on and advances in it will help us build good motor primitives taking the visuomotor feedback loop into account.

However, the readers here (people who know RL, DL) are not laymen on Hacker News / Reddit / casual readers of MIT Technology Review. So, we will not tolerate arbitrarily grandiose and incorrect statements made to portray your paper as a big advance. It is better to be grounded and stay true to the field as far as the writing on the paper is concerned.

(i) "There has been a lot of recent progress in the field of Reinforcement Learning (RL). However, most of the successful applications have been confined to the artificial world of video games (Mnih et al., 2015b) or simulations (Lillicrap et al., 2016)."
   Not true. There have been successful applications for reducing power consumption in data centers, recommendations, advertisements, optimizing commercial applications for good user experience. Reinforcement Learning has existed even before Vlad Mnih invented DQN. There is a whole history of work on bandits and how to use them for online learning, optimization for ads, recommendations in e-commerce, etc. AutoML is a new upcoming application of reinforcement learning. This statement (the very first couple of sentences in the paper) clearly tells that the authors have no idea what they are talking about.
(ii) "While the field of RL was born out of need to make our robots learn how to perform actions, none of the recent advances have translated to success in the field of robotics. Why is that?"
   This is again such an erroneous version of the field casually made with no evidence. Reinforcement Learning was studied from the perspective of behavior psychology initially. There is no evidence that it started only for making robots learn. One more evidence that the authors don't know what they say. Secondly, what classifies as success as far as robotics is concerned? As far as I know, there have been impressive demonstrations of reinforcement learning on real robots, such as OpenAI's Dactyl, Sergey Levine's Grasping work at Google, etc. So, are you saying that those are insignificant successes?

---

> ### Public Comment · (anonymous) · 2018-11-05
> **Continuation**
>
> (iii) "In model-free Reinforcement Learning(RL) paradigm, the robot will try and try until it is able to stack objects and once it hits a successful instance, it is used as a positive signal (‘reward’) to promote these policy parameters. How does the robot try? Due to lack of any other signals from the environment, most-often robots use random-exploration policies (or random trajectories). It is clear that if the rewards are sparse, it may take millions of random “tries” before it hits any success. Clearly this approach is only scalable in video-games and not real-world robotics."
>    Not correct. People have demonstrated that with parallel data collection with a farm of robots (Levine's work) / good sim2real transfer (OpenAI Dactyl), model-free reinforcement learning indeed scales even for sparse rewards. So, one more evidence that the authors don't understand the notion of 'scalable' but use it freely with no qualms.
> (iv) "Another possibility is to use model-driven approaches. Here, the robot will learn a model of the world from our millions of interactions and use the model to simulate and search. But what millions of interactions should be performed to build our models? Again due to lack of any external information, the most common approach is using random interactions to explore and build the world model (Agrawal et al., 2016; Levine et al., 2016; Pinto et al., 2016). Building a good model will require enormous number of interactions."
>     The notion of model is completely incorrect here. None of the mentioned citations above actually build a world model. Agrawal learns inverse models, while Levine and Pinto learn a grasp success predictor. Agrawal's approach doesn't allow you to "simulate". It is a greedy one-step planner continuously executed in the real world till the provided goal is achieved. Levine uses cross-entropy method to adjust the actions for optimizing the output of the grasp success predictor. That's just basic policy optimization in a derivative-free manner and the whole pipeline is effectively a recurrent model-free approach (success predictor being the equivalent of a value function). The authors should clearly clarify that they are talking about models at the pixel level here, because there are lots of successes with reduced number of interactions by learning state-level models (Kurutach et al, Nagabandi et al, Chua et al).
> (v) "it acts as reward shaping function which improves sample efficiency but not by a significant amount." - What will be significant enough for you? It is worth clarifying that to the readers. Random Network Distillation by OpenAI surpassed human level on Montezuma Revenge. Is that insignificant? Considering this paper was submitted even before that, I am curious if you consider all prior successes in exploration insignificant. Count based exploration could unlock upto 20+ rooms on Montezuma.

---

> > ### Public Comment · (anonymous) · 2018-11-05
> > **Continuation**
> >
> > (vi) "Another possibility is to use the inspiration from humans: humans even try to explore the world without the context of task. ...."
> >    Why is this "another possibility"? Burda et al's work falls into the same category as intrinsic rewards. They do an extensive evaluation of different intrinsic reward schemes. "Task agnostic" makes no sense here. When there are no extrinsic rewards provided,  the policy is inherently task-agnostic.
> > (vii) "Our paper investigates exploration from an optimization viewpoint and asks a simple question: can we formulate curiosity reward as a differentiable function? We believe a differentiable reward function would enable to us to be sample efficient and for the first time ever, implement exploration on a real-world physical robot."
> >   Factually incorrect. Any reward function that is a DNN function of (s,a) is differentiable. For example, a value function built on a density model over states and actions. Or a simple model that takes in (s,a) and predicts a scalar. Curiosity reward being differentiable or not is hardly the issue. The point is you can make it differentiable function in multiple ways.
> >    Secondly, this paper is NOT the first to show exploration on a real-world physical robot. Oudeyer's lab has a lot of work in this direction. Example: https://www.youtube.com/watch?v=NOLAwD4ZTW0  - Intrinsically Motivated Goal Exploration Processes with Automatic Curriculum Learning": https://arxiv.org/abs/1708.02190
> > (viii) "The standard way to optimize policy to maximize sequence of intrinsic rewards is to either use REINFORCE (i.e., on-policy way) or regress to learn value estimates (i.e., off-policy) as discussed in the previous section. Both these approaches do not make any use of the structure present in the design of rit. While these are unbiased estimators for training policy parameters with respect to rit, they suffer from high variance which is a known issue in reinforcement learning and an active area of research (Schulman et al., 2015). For instance, REINFORCE roughly amounts to saying that the agent should change the probability of the executed action in proportion to rewards received which fluctuates with the reward trajectories, leading to high variance. It gives no signal as to what action to take if the current action did not lead to a good reward."
> >     Not correct. There has been more recent work on using control variates to structure the critic to take into account the action executed and learn action-dependent baselines which could be estimated off-policy. They do not suffer from the variance issue as REINFORCE does. Secondly, Bengio's straight-through estimator doesn't suffer from the variance issue.
> > (ix) The term differentiable exploration doesn't really mean anything. It doesn't solve the exploration issue any better than optimizing a policy on an intrinsic reward that is computed on environment transitions in different ways. The true exploration (which is intractable) is to maximize state visitation frequency. You resort to using straight through anyway. Calling straight through differentiable amounts to calling biased gradient estimators differentiable. Why invent new names for things that already exist?
> > (x) "We perform simulation experiments to help us setup the right parameters and do extensive comparisons against REINFORCE and DQN" - These baselines are absolutely poor and as good as random policies. It is clearly not executed well. Makes me wonder the point of showing the comparisons.

---

> > > ### Author Response · Authors · 2018-11-05
> > > **[1/3][Author's Response] The Disconnect: Is 100yr training sample-efficient or scalable?**
> > >
> > > We would like to first thank the commentator for the comments. These comments (and our reply below) definitely help us show the disconnect between two communities; the two spectrums of research directions. For example,
> > >
> > > ==> Commentator: "good sim2real transfer (OpenAI Dactyl), model-free reinforcement learning indeed scales even for sparse rewards."
> > > ==> Us: 100 years of experience for just manipulating one type of cube is completely unscalable. Tens-of-thousands of tasks each requiring tens of years of experience (even if experience requirements are diminishing) does not seem will get us far in AI.
> > >
> > > ==> Commentator: "there have been impressive demonstrations of reinforcement learning on real robots, such as OpenAI's Dactyl, Sergey Levine's Grasping work at Google, etc."
> > > ==> Us: Robotics is still missing its aha ImageNet moment (which computer vision had) and therefore a lot of results remain elusive and kind of non-replicable. That is why we still do not see every robotics person flocking to RL. Classical robotics can get some of these results with no training (https://www.youtube.com/watch?v=KVGn8tP9klI)
> > >
> > > Which version is true? Probably both or probably none. This paper and our introduction hoped to tell the story from the other side. We want to emphasize our intention is not to grandify the claims but to highlight the disconnect between communities and we will be happy to soften the tone (especially in one place where we kind of agree with the commentator we should soften). Again these are changes in writing style in one paragraph or line and do not change the story or the experimental claims of the paper.
> > >
> > > We believe research communities should be open (and respectful) enough to hear multiple perspectives and see how things will unfold in long-term. It is not healthy to say things like: "authors don't understand the notion of scalable" when things are not really as black/white. For the commentator, 100 years for one cube or billions of frames for one game seems scalable but for us, this highlights how sample-inefficient these algorithms are!
> > >
> > > Finally, we highlight that commentator’s comments like action-dependent baselines "do not suffer from the variance issues" are not true and we highlight below the papers that clearly show these problems still exist.
> > >
> > > We answer all individual comments below point-by-point.

---

> > > > ### Author Response · Authors · 2018-11-05
> > > > **[2/3][Author's Response] Does RL no longer have variance issues?**
> > > >
> > > > We first highlight technical and experimental comments because they are critical for evaluation of paper and then we will answer stylistic/philosophical questions:
> > > >
> > > > (RE: viii) Commentator: "RL with action-dependent baselines does not suffer from the variance issue as REINFORCE does."
> > > > There have been great advances in RL optimization. However, the community still believes that the variance issue in RL is an open research area and it would be an over-claim to say that variance issue in RL no longer exists. For instance, in particular, see [Tucker et.al., ICLRW 2018] from Levine’s group which shows that "learned state-action-dependent baselines do not in fact reduce variance over a state-dependent baseline" (quote from that paper). Therefore, we believe our comments regarding variance are correct.
> > > >
> > > > (RE: ix) Here, differentiability is emphasized with respect to the actual (intrinsic) reward function agent is optimizing for. This is almost always is treated as a "scalar" quantity (black-box) in RL-based optimization (see Section-2 and Section-6). We do not argue that the use of straight-through estimator is being differentiable. Moreover, note that one would need to use straight through only in discrete action space, and directly back-prop in continuous cases.
> > > >
> > > > (RE: x) It is not scalable to tune REINFORCE or DQN baselines on the real world. Hence, we tune them in the simulation. For instance, see Figure-4 in the appendix, which validates that the same REINFORCE baseline implementation works well in presence of hard-coded external reward, but not well during exploration. We used the same hyper-parameters for our method by only changing the optimization objective. So, to reiterate for both approaches parameters were optimized in simulation and then same parameters were used in real-world experiments!
> > > >
> > > > (RE: vii) Our main argument is that intrinsic rewards have a lot of structure which gets lost if we learned to estimate it just as a "scalar" quantity, which is essentially the same as treating rewards as black-box. Policy gradients, value functions, Q-learning follow this regime which is differentiable, but the reward structure gets already lost in learning them. This is described in detail in Section-2.
> > > > The work of [Forestier, Mollard, Oudeyer, 2017] on real-robot is using low-dimensional fully-observable state-space information (e.g. exact location of objects etc.) and dynamic motion primitives to lower action complexity. This is in contrast to our setup, where our robotic agents learn to explore using high-dimensional raw images with large action-space beginning without any notions of objects or their existence. Our exploration learns the concept of an object directly from scratch rather than being hand-coded. Furthermore, the setting of Oudeyer is using the same object and does not try on a different set of objects and hence it is not clear how generalizable it is.

---

> > > > > ### Author Response · Authors · 2018-11-05
> > > > > **[3/3][Author's Response] Does DeepRL for robotics work?**
> > > > >
> > > > > We now answer philosophical/stylistic comments:
> > > > >
> > > > > (RE: i) We highlighted "most" recent applications of RL being on games/simulation (we never used ONLY or ALL).  The applications pointed by the commentator (e.g., recommendations, advertisement etc.) are mostly that of the ‘multi-arm bandits’. However, multi-arm bandits are not the same as reinforcement learning and instead mostly deal with "stateless" problems where actions taken do not affect the next state. They are a much-restricted class of problems and have been used for advertisements in e-commerce. They are not applicable to the real-world robotics setting where an agent acts in a real manipulation environment.
> > > > >
> > > > > (RE: ii) We will soften the point to say one of the major RL applications was supposed to be Robotics. That said, we still do not believe RL is as widely accepted as ConvNets in computer vision. Most robotics researchers still do not use robot learning or believe its results are as impressive as the commentator seems to suggest (robotics researchers are not flocking to RL). We believe there is long-way before RL can be applied to real-world robotics applications.
> > > > >
> > > > > (RE: iii) As mentioned above, definitions of scalability can have a different perspective. 100-years of experience for a single task of manipulating cubes is unscalable for us. Consider tens-of-thousands of tasks each requiring years of experience does not seem the scalable way of doing things (even if experience requirements are diminishing). The parallel way of data-collection works only in primitive tasks like grasping and poking where success rates for random are high as well.
> > > > >
> > > > > (RE: iv) Our point is indeed further emphasized by the commentator’s remark. There is very little to no work which builds actual accurate, long-term, non-greedy models from raw sensory observations in real robotics setting. We will clarify that such models should operate on raw sensory observations (e.g. pixels), and not state-space because estimating the exact state-space of the environment is a big research area in itself (i.e., system identification), and still an unsolved problem (i.e., estimating exact position of objects in a scene in computer vision).
> > > > >
> > > > > (RE: vi) Most of the classical work on learning with intrinsic rewards has been to just augment an external reward, in which case the learned policy is specific to the task it is trained for. An "alternate" way here refers to that of developmental psychologist where intrinsic motivation itself (i.e., without any external reward or end-goal) is seen as the primary driver in the early stages of development to learn skills which are useful in future. [Burda et.al. 2018] is one such example where interesting behaviors emerge by only training for intrinsic rewards across 54 simulated environments.
> > > > >
> > > > > (RE: v) Yes, Random Network Distillation (RND) paper showed up just 6 days ago on arXiv, much after the submission deadline. This paper also shows results on video games. While this paper shows great results on playing video games, but note that it uses "a total of 1.97 billion frames of experience" [quote from RND paper] for each experiment! This is not at all scalable to real-robotic settings, even in a robotic arm-farm setup.
> > > > > Again we emphasize our argument is that these RL results are significant and great advances but just not scalable to learning with robots.

---

### Public Comment · ~Aravind_Srinivas1 · 2018-11-06
**Good engineering aspects worth appreciating**

I read the spicy discussion below (very Reddit like). Though the anonymous comment highlighted some issues in the writing, I think the comment fails to see some details in this paper that are really worth appreciating. As someone who has worked a bit in visual continuous control, I find some details refreshingly fresh:

(1) Going for a very high dimensional discrete binned action space: This is a really good idea and might go on to become a standard in applying deep learning for robotics. Policies need more signal to learn better and faster and discrete has in general worked out better in deep learning.

(2) Learning from high-resolution inputs: For some reason, applications of RL to robot control have tried to use 84x84 (or 64x64) inputs inspired from the original DQN architecture so far.  Learning from more high-resolution inputs makes more sense.

(3) The rotation decision by augmenting the input: This is a nice idea taking the geometric aspect into consideration.

That said, it is possible that the DQN and REINFORCE baselines don't really work because of the high dimensional action space (especially REINFORCE). With a more engineered action space, they could. But this warrants more investigation in the future.

---

### Author Response · Authors · 2018-11-20
**[Author's Common Response] Added New Results; Will update Intro; Explaining Objective**

We thank reviewers for their time and useful feedback. It is always the authors' responsibility that the readers understand the context, the underlying technical approach, and the experiments. We believe we were short in all the three, but we also believe most of the content is there and all it needs is rewriting. On top of that, to convince reviewers we have also performed additional experiments including in simulation to demonstrate our baselines are the baselines reviewers are asking for (we just did not present them properly). Therefore, we urge both reviewers and AC to give us a second chance and consider the paper in light of the rebuttal.

Experiments -- REINFORCE baseline is the curiosity baseline [Pathak et al. 2017]
==========
First, we would handle the experimental concerns because we agree if we cannot demonstrate empirically that our approach works, then the paper should not be accepted. Reviewers are concerned we are comparing with DQN and REINFORCE but not with other exploration approaches. We would like to point out that both DQN and REINFORCE are not using external rewards and hence they are exploration approaches. Specifically, REINFORCE baseline is THE curiosity baseline and DQN is another optimization approach used for the same objective as curiosity baseline [Pathak et al. 2017]. One minor difference in the use of curiosity [Pathak et al. 2017] for our experiments is the use of pre-trained ImageNet feature. This feature is stationary which further helps the curiosity formulation as discussed in the follow-up [Burda et. al. 2018].

In order to further demonstrate that our baselines are credible, we took the implementation and curves from recent curiosity paper [Burda et. al. 2018]. We use the curiosity baseline provided in the paper and to demonstrate our objective function for exploration is meaningful, we add our objective to the original exploration objective. Note both these environments require long-term modeling and since using current approaches long-term models are hard to learn; we use our objective in conjunction with existing models. On both the environments, our objective seems to provide gain in efficiency. Hopefully, this confirms that the curiosity baselines used in the paper (referred to as REINFORCE) is a meaningful baseline. Please see results in Appendix Section-C of the updated paper.

Context
==========
We agree that some of our context placement was too strong, and we would be happy to soften it. But to put our paper in the right context. First, we agree that current Deep-RL approaches have made significant advances but as most RL experts would agree: sample efficiency still is a huge bottleneck. Again almost all RL experts would agree, this has been one of the bottlenecks towards bringing RL to robotics (without using expert demonstrations).
One possible way is to train in simulation and then transfer to real-world robots; this might work but currently, in object manipulation, it is hard to transfer due to reality gap (contact dynamics are hard to simulate). As a research community, we should also investigate the alternative way: to train from scratch on the robot in real-world settings. This is where our paper comes in. Current RL-based exploration approaches (e.g., curiosity and visitation counts) use reward functions that are black-box in nature. Hence the gradients are not that meaningful which reduces the sample complexity. In our paper, we propose a new exploration objective that can be optimized via supervised learning (instead of reinforcement learning). We do not claim that this new objective has all properties of the previous one, and we elaborate on this below.

Technical Insights -- What does this new objective actually do?
===========
AR2 rightly asks the question: what does this new objective metric actually do? To answer this, and give intuition about the objective we have added a new section to the paper, see Appendix Section-D. We will move some part of it to the main paper in the final version. To summarize, due to the differentiable nature of our objective and presence of local constancy approximation, our objective does not behave identically to the original curiosity reward which is optimized by policy gradients (REINFORCE). The main difference in behavior is in case of interactions which have the same repetitive outcome: our method explicitly forces towards the action with different outcomes, while original curiosity objective penalizes the actions for repetitive outcomes. We further add pointers to the Experimental-Design in convex optimization literature which bears a close similarity to our setup. Please refer to Section-D for details.

We now answer other individual comments in the direct replies to reviewers.

---

### Meta-Review · Area_Chair1 · 2018-12-13

**Confidence:** 5
**Recommendation:** Reject

**Metareview:**

The authors propose implementing intrinsic motivation as a differentiable supervised loss coming from the error of a forward model, rather than the black box style of curiosity reward. The motivation is that this approach will lead to more sample efficient exploration for real robots. The use of a differentiable loss for policy optimization is interesting and has some novelty. However, the reviewers were unanimous in their criticism of the paper for poor baselines, unclear experiments and results, and unsupported claims. Even after substantial revisions to the paper, the AC and reviewers were unconvinced of the basic claims of the paper.